# SPARC Induces E-Cadherin Repression and Enhances Cell Migration through Integrin αvβ3 and the Transcription Factor ZEB1 in Prostate Cancer Cells

**DOI:** 10.3390/ijms23115874

**Published:** 2022-05-24

**Authors:** Fernanda López-Moncada, María José Torres, Boris Lavanderos, Oscar Cerda, Enrique A. Castellón, Héctor R. Contreras

**Affiliations:** 1Laboratory of Cellular and Molecular Oncology (LOCYM), Department of Basic and Clinical Oncology, Faculty of Medicine, University of Chile, Santiago 8380453, Chile; fernanda.lopez.m@gmail.com (F.L.-M.); martorres@ug.uchile.cl (M.J.T.); 2School of Medical Technology, Austral University of Chile, Puerto Montt 5504335, Chile; 3Millennium Nucleus of Ion Channel-Associated Diseases (MiNICAD), Santiago 8380453, Chile; lavanderos.boris@gmail.com (B.L.); oscarcerda@uchile.cl (O.C.); 4Program of Cellular and Molecular Biology, Institute of Biomedical Sciences, Faculty of Medicine, Universidad de Chile, Santiago 8380453, Chile

**Keywords:** SPARC, cadherins, epithelial–mesenchymal transition, integrins, zinc finger E-box-binding homeobox, prostate cancer

## Abstract

Secreted protein acidic and rich in cysteine (SPARC), or osteonectin, is a matricellular protein that modulates interactions between cells and their microenvironment. SPARC is expressed during extracellular matrix remodeling and is abundant in bone marrow and high-grade prostate cancer (PCa). In PCa, SPARC induces changes associated with epithelial–mesenchymal transition (EMT), enhancing migration and invasion and increasing the expression of EMT transcriptional factor Zinc finger E-box-binding homeobox 1 (ZEB1), but not Zinc finger protein SNAI1 (Snail) or Zinc finger protein SNAI2 (Slug). It is unknown whether the SPARC-induced downregulation of E-cadherin in PCa cells depends on ZEB1. Several integrins are mediators of SPARC effects in cancer cells. Because integrin signaling can induce EMT programs, we hypothesize that SPARC induces E-cadherin repression through the activation of integrins and ZEB1. Through stable knockdown and the overexpression of SPARC in PCa cells, we demonstrate that SPARC downregulates E-cadherin and increases vimentin, ZEB1, and integrin β3 expression. Knocking down SPARC in PCa cells decreases the tyrosine-925 phosphorylation of FAK and impairs focal adhesion formation. Blocking integrin αvβ3 and silencing ZEB1 revert both the SPARC-induced downregulation of E-cadherin and cell migration enhancement. We conclude that SPARC induces E-cadherin repression and enhances PCa cell migration through the integrin αvβ3/ZEB1 signaling pathway.

## 1. Introduction

Prostate cancer (PCa) is one of the most frequently occurring cancers in men worldwide. Regardless of advances in diagnosis and treatment, metastasis is still the greatest challenge for PCa survival. While patients with localized disease have a 5-year survival of 99%, patients with metastatic disease have only 28% [1,2]. Bone tissue is the main site of metastasis for PCa. Autopsy studies estimate that approximately 70% of patients who die from PCa present bone metastases [3]. The dissemination of PCa cells to bone and other tissues requires enhanced invasive and motile capacity of the tumor cells to enter the circulation and reach distant organs. In most carcinomas, this progression depends on the activation of the epithelial–mesenchymal transition (EMT) program in the neoplastic cells. The EMT is a trans-differentiation program characterized by the loss of epithelial features, including epithelial cell–cell contact molecules and apicobasal polarity, and the gain of mesenchymal molecular markers such as front-rear polarity and enhanced migration and invasion [4]. The EMT program has long been described as a binary process with two opposing cell populations: epithelial cells and mesenchymal cells. The latter are frequently defined by loss of E-cadherin and increased vimentin expression. However, in recent years, multiple studies indicate that the EMT process occurs gradually, characterized by several intermediate cellular states, with different levels of expression (and often co-expression) of epithelial and mesenchymal markers, presenting intermediate molecular, morphological and functional characteristics between both cell types [5,6,7,8]. Although all the subpopulations present during EMT (epithelial, early EMT, intermediate EMT, late EMT, and complete mesenchymal) can show some degree of plasticity, the evidence suggests that hybrid or intermediate EMT populations are those showing the highest degree of plasticity and clonogenic capacity which, under in vivo conditions, can give rise to tumors with different subpopulations [5,8]. Consistent with this, recent results from our laboratory show that PC3 cells with silenced SPARC have lower clonogenic capacity, lower ability to form prostatospheres under non-adherent culture conditions, as well as a lower expression of the CD44 marker and the transcriptional factor SOX2 [9].

In PCa and other types of cancer, secreted protein acidic and rich in cysteine (SPARC) has been associated with the induction of EMT-like features, such as E-cadherin loss and enhanced migration [10,11,12]. SPARC, also known as osteonectin and basement membrane-40 (BM-40), is a matricellular glycoprotein that modulates interactions between cells and their surrounding microenvironment [13]. In the bone tissue, SPARC is highly expressed by osteoblasts, which promote the formation, maintenance, and repair of bone and regulate procollagen processing and matrix assembly and mineralization [14]. Experiments in PCa cells have shown that both bone extracts and purified SPARC act as chemoattractant factors that enhance migration and invasion [15,16].

Moreover, SPARC is also expressed by tumor cells. For example, SPARC is highly expressed in PCa cell lines derived from bone metastasis, such as PC3 and V-CAP [15]. Moreover, its expression has also been found in tumor cells from biopsies of primary tumors, with higher expression in poorly differentiated PCa [11,17]. Recently, we described that PCa cells overexpressing SPARC have a lower expression of prostate epithelial markers such as E-cadherin and cytokeratin 18 and an increased expression of the mesenchymal marker vimentin. Importantly, SPARC also increases the expression of the transcription factor Zinc finger E-box-binding homeobox 1 (ZEB1), but not other EMT transcription factors such as Zinc finger protein SNAI1 (Snail) or Zinc finger protein SNAI2 (Slug) in PCa cells [11]. ZEB1 is considered one of the master genes of the EMT, inducing EMT in epithelial cells through direct E-cadherin transcriptional repression and other components of epithelial junctions such as occludins, claudins, and desmoplakins. Furthermore, ZEB1 activates the expression of mesenchymal genes such as N-cadherin, vimentin, and several matrix metalloproteinases [18]. However, it is not known whether the SPARC-induced downregulation of E-cadherin in PCa cells depends on ZEB1. 

On the other hand, several integrins have been described as mediators of some of the SPARC effects on cancer cells [15,19,20]. Integrins are transmembrane cell adhesion molecules formed by the heterodimerization of alpha and beta chains. Integrin signaling depends on the formation of cell adhesion complexes including components such as focal adhesion kinase (FAK) and Src; adaptor proteins such as talin, paxillin, and vinculin; and GTPases from the Rho family, modulating cytoskeleton remodeling and focal adhesion turnover [21,22]. Studies in cancer have shown that integrin signaling can mediate the activation of the EMT program induced by other factors present in the tumor microenvironment, such as the epidermal growth factor (EGF), the fibroblast growth factor (FGF), and the pituitary tumor-transforming gene (PTTG) [23,24,25]. Considering this background, we hypothesize that SPARC induces E-cadherin repression through the activation of integrins and ZEB1.

## 2. Results

### 2.1. SPARC Downregulates E-Cadherin and Enhances ZEB1 and Vimentin Expression in PCa Cell Lines

In previous work, we have described that the DU145 PCa cell line expresses low levels of E-cadherin in comparison to other prostate cell lines. To evaluate whether targeting SPARC can recover E-cadherin expression in these cells, DU145 cells were stably transduced with two different short hairpin RNAs against SPARC (shSPARC). The results of SPARC silencing demonstrated that targeting SPARC increases the expression of E-cadherin and decreases the expression of the transcription factor ZEB1, a direct repressor of E-cadherin (Figure 1A–C). Moreover, we have previously demonstrated that the overexpression of SPARC in LNCaP cells decreases the expression of E-cadherin and increases the expression of vimentin. To evaluate whether recombinant SPARC could exert the same effect, LNCaP cells were incubated with different concentrations of human SPARC for 6 hrs. The RT-qPCR analysis showed that incubation with 0.5 or 1 μg/mL of SPARC decreases the expression of E-cadherin mRNA, showing similar effects to the ectopic SPARC expression by lentiviral transduction (SPARC_HA) (Figure 1D). To demonstrate that SPARC also decreases E-cadherin protein, LNCaP cells were incubated with 1 μg/mL of SPARC and the expression of E-cadherin and vimentin were evaluated by Western blot for up to 72 h. Accordingly, in line with our previous observation, the incubation with recombinant SPARC induced a transient decrease in E-cadherin and an increase in vimentin, with a peak at 24 h (Figure 1E–G).

### 2.2. SPARC Increases the Expression of Integrin β3 Subunit and Requires Its Activity to Inhibit E-Cadherin

To evaluate whether the expression of some integrin subunits was modified by SPARC in PCa cells, the levels of the integrin subunits αv, α5, β1, β3, β4, and β5 were evaluated through Western blot in PC3 cells with the stable knockdown of SPARC (shSPARC), and LNCaP cells with stable SPARC overexpression (SPARC_HA). Both the silencing and overexpression of SPARC modified the expression of several integrin subunits. Interestingly, among the studied integrins, subunit β3 was only downregulated when SPARC was silenced and upregulated when SPARC was overexpressed (Figure 2A–C).

The integrin β3 subunit heterodimerizes mainly with the integrin αv subunit; for this reason, the basal levels of expression of these two subunits were analyzed in different PCa cell lines. The expression of both subunits was determined by RT-qPCR and Western blot, in the 22Rv1, LNCaP, PC3, and DU145 cell lines. Compared to the 22Rv1, LNCaP, and DU145 cell lines, the PC3 cells had the highest protein expression of the integrin αv, the β3 subunits and SPARC (Figure 3A–C). 

To evaluate whether the activity of integrin αvβ3 was necessary for SPARC-induced E-cadherin downregulation, LNCaP cells overexpressing SPARC were incubated with an RGD peptide that specifically blocks the activation of integrin αvβ3. As shown previously, overexpressing SPARC in PCa cells decreases the expression of E-cadherin and increases the expression of vimentin. However, blocking integrin αvβ3 reverts this effect, increasing E-cadherin and decreasing vimentin expression (Figure 3D,E).

### 2.3. Silencing of SPARC Decreases the Phosphorylation of FAK-Y925 and Impairs Focal Adhesion Formation

Because focal adhesion kinase (FAK) mediates integrin αvβ3 signaling, the effect of knocking down SPARC on the expression or phosphorylation status of FAK was evaluated. As shown in Figure 4A,B, no changes were observed in the expression of total FAK when SPARC was silenced. However, there was an increase in phosphorylation in the Y397 residue and a decrease in phosphorylation in the Y925 residue. Moreover, knocking down SPARC decreased the mRNA of Rac Family Small GTPase 1 (RAC1), Ras Homolog Family Member A (RHOA), and the Integrin Linked Kinase (ILK). 

The 925-tyrosine residue in the FAK focal adhesion targeting region is a key event for the focal adhesion’s formation. Therefore, the focal adhesions of ShScramble and ShSPARC PC3 cells were compared (Figure 4D). An increase in the number of focal adhesions per cell was observed in ShSPARC PC3 cells compared to ShScramble PC3 cells (Figure 4E). However, these focal adhesions were smaller in cells with SPARC silencing compared to the focal adhesions formed in the control PC3 cells (Figure 4F).

### 2.4. Integrin αvβ3 Activation and ZEB1 Expression Are Required for SPARC-Induced E-Cadherin Downregulation and Enhanced Migration

To evaluate whether the expression of ZEB1, together with the activity of the integrin αvβ3 were necessary for the SPARC-induced decrease in E-cadherin, ZEB1 was knocked down in LNCaP cells using lentiviral particles containing a specific ShRNA (LNCaP ShZEB1). As shown in Figure 5A,B, ShZEB1-LNCaP cells showed a lower expression of ZEB1 compared to the LNCaP cells transduced with a non-specific shRNA (ShScr-LNCaP). To evaluate whether this silencing was functional, the expression of E-cadherin was determined, which is directly repressed by ZEB1. As expected, the silencing of ZEB1 increases the expression of E-cadherin. Interestingly, the silencing of ZEB1 also decreases the expression of SPARC (Figure 5A,B). Then, the LNCaP control cells and ShZEB1-LNCaP knocked down cells were stimulated with SPARC, and the integrin αvβ3 activity was inhibited with an RGD peptide. In the LNCaP control cells, we observed that SPARC decreased the expression of E-cadherin, an effect that was prevented when the activation of integrin αvβ3 was blocked (Figure 5C). However, in ZEB1-silenced LNCaP cells, SPARC showed no effect on E-cadherin expression, regardless of integrin blocking (Figure 5D), showing that ZEB1 expression and integrin αvβ3 activation are necessary for SPARC-induced E-cadherin downregulation. 

Finally, to assess whether integrin αvβ3 and ZEB1 mediate SPARC-induced migration, a transwell migration assay and a wound healing assay were performed. In the LNCaP control cells, SPARC increased the motility and transmigration capacities of the LNCaP cells, an effect that was completely reversed by blocking integrin αvβ3 (Figure 5E,G,I). On the other hand, in the ZEB1-knockdown LNCaP cells, SPARC showed no effect on the motility and transmigration capacities of these cells, although blocking integrin αvβ3 decreased the transmigration capacities of the LNCaP cells (Figure 5F,H,J). In each experiment (Figure 5E,F), the results were normalized with an arbitrary value assigned to the control.

## 3. Discussion

One of the identity signals of the EMT in cancer is the loss of functional E-cadherin [4]. In PCa there is a strong association between the loss of E-cadherin and tumor undifferentiation, measured as Gleason score [26]. In previous work, we have described that SPARC is overexpressed in high Gleason score PCa tissue and induces changes associated with the EMT [11]. Indeed, the effects of SPARC on EMT showed that silencing SPARC induces morphological, molecular and functional changes related to EMT. For example, we observed that knocking down SPARC induces changes in cell polarity, increases the expression of E-cadherin, and decreases the expression of ZEB1, N-cadherin, and vimentin, while decreasing the invasive and migratory capacity of PCa cells, as evaluated by the wound-healing and transmigration assays [11]. Therefore, we aimed to further evaluate the relationship between SPARC and the loss of E-cadherin in PCa. E-cadherin, coded by the CDH1 gene, is a cell–cell adhesion molecule that maintains the functional characteristics and integrity of the epithelia, forming complexes with the actin cytoskeleton via cytoplasmic catenins. The loss of E-cadherin expression disrupts this complex resulting in the loss of cell polarity, epithelial denudation, and increased epithelial permeability in a variety of tissues, promoting cancer cell migration, invasion, and metastasis [27,28,29] events that are also induced by SPARC in PCa cells [15,16,26]. This is interesting because it has been shown that the loss of E-cadherin in mouse prostatic luminal epithelial cells has been associated with the development of prostatic intraepithelial neoplasia (PIN), a precursor lesion of PCa [29]. This evidence reported in the literature suggests that SPARC could be contributing not only to the dissemination of PCa but also to its development.

SPARC-induced cell migration appears to be mediated by different cell–matrix integrins. For example, De et al. describe that migration towards SPARC requires the activation of integrin αvβ3 and αvβ5 [18], while Girotti et al. describe that the invasion of melanoma cells stimulated by SPARC depends on integrin α2β1 [15]. In our PCa model of SPARC overexpression and silencing, SPARC regulated the expression of several integrin subunits. Among them, the integrin β3 subunit presented changes concordant with the expression of SPARC. It has been reported that the ectopic expression of integrin β3 induces the expression of SPARC in melanoma cells [30] and that the β4 subunit regulates the expression of SPARC in breast cancer cells [31]. This evidence, together with our findings in the wound healing assay showing that integrin αvβ3 and ZEB1 mediate SPARC-induced migration, suggests that SPARC and some integrin genes could be positively and reciprocally regulated, contributing to the migration and invasion of cancer cells. Although RGD, used in this study, also blocks some others integrin subtypes, this peptide blocks αvβ3 with high affinity and selectivity. On the other hand, our result on ZEB1 knockdown (using a specific shRNA) suggests that this integrin could be the main candidate, according to the literature and our data. Furthermore, the increase in αvβ3 integrin by SPARC could facilitate the metastatic process, particularly bone metastases, which is the main metastatic niche of PCa [32]. Some evidence shows that the overexpression of αvβ3 integrin increases the number and size of bone metastases from ovarian cancer [33], in addition to increasing the migration of cancer cells towards the bone sialoprotein, which is highly expressed in the bone stroma, similar to SPARC [34,35,36].

On the other hand, in SPARC-silenced PC3 cells, the decrease in the expression of integrin αvβ3 was accompanied by a decrease in the expression of genes related to migration and integrin activity, specifically RAC1, RHOA, and ILK. Shi et al. have previously reported a decrease in ILK activity and FAK Y397 phosphorylation in glioma cells with SPARC silencing [37]. However, in the PCa cells with silenced SPARC, an increase in P-FAK Y397 and a decrease in P-FAK Y925 were observed. The phosphorylation of FAK Y925 is key for focal adhesion turnover because it prevents the paxillin from binding to the focal adhesion targeting (FAT) domain [38]. Paxillin regulates cytoplasmic extension and cell migration [39], which could explain why both processes are affected in SPARC-silenced PC3 cells. SPARC silencing alters the focal adhesion formation, increasing the number and decreasing the size compared to the control cells, which is consistent with a lower migratory capacity [40].

Furthermore, we observed that αvβ3 integrin mediates the SPARC-induced downregulation of E-cadherin. Integrin αvβ3 mediates EMT induction by several factors, such as transforming growth factor-beta 1 (TGF-β1), pituitary tumor transforming gene (PTTG), and fibroblast growth factor 1 (FGF1) [23,24,41,42,43]. On the other hand, at the intracellular level, ZEB1 increased expression and the consequent E-cadherin repression could be mediated by the activation of ILK and FAK. However, it would be necessary to carry out additional experiments to elucidate whether, in this case, the activation of ZEB1 depends on ILK, FAK, or both. In lung cancer cells, there is evidence that FAK mediates PTTG-induced EMT [24] whereas, in mammary cells, ILK function is required for TGF-β1-induced EMT [44]. However, this study does not determine which of these pathways are involved. Indeed, in previous studies by our group, we have observed that the modification of SPARC expression is accompanied by significant changes in ZEB1 in the same direction (ZEB1 increases when SPARC is overexpressed and decreases when SPARC is silenced). However, no similar effect was observed in the other EMT-inducing transcriptional factors, such as Snail and Slug. However, other authors have reported that integrin signaling can induce Snail and Slug expression in other cancer models, for example, ILK increases Snail expression in non-small cell lung cancer [45], while integrin αvβ3 induces Slug in breast cancer with effects linked to stemness [46]. Therefore, it could not be ruled out that some of the observed effects were tumor-dependent.

The increase in ZEB1 by SPARC and its implication as a mediator of E-cadherin inhibition raises many questions regarding other cellular processes that SPARC could be regulating through ZEB1. We have previously described the effect of SPARC on the expression of the transcriptional factor ZEB1, finding an increase in its expression in cells with SPARC overexpression and a decrease when SPARC was knocked down [11]. Consistent with our previous work, we found an increase in ZEB1 expression in LNCAP cells treated with 1 ug/mL SPARC at 24 and 48 h (data not shown). 

We used different cell lines depending on the experiment. This was designed to show clearer results so that we could choose the cell lines with adequate expressions of markers to be studied and genes to be silenced. This is a limitation of this study, since not all of our results are shown in a panel of cell lines. It would be necessary to select at least two or three cell types with similar expression profiles of markers for further study.

There is growing evidence that ZEB1 not only plays an important role in the EMT process, but can also control critical cellular functions including differentiation, proliferation, response to cell damage, and survival [18]. For example, ZEB1 could be relevant in the early tumorigenesis of pancreatic carcinoma [47,48], and in non-epithelial tumors such as melanoma [49]. Moreover, ZEB1 has been associated with the development of the cancer stem cell (CSC) phenotype [50,51] through the regulation of miRNAs controlling the expression of stemness transcription factors such as SOX2 and KLF4 [52]. Finally, ZEB1 has also been associated with high resistance to therapy in various models. For example, ZEB1 is involved in PCa resistance to docetaxel [53] and, in ovarian cancer, resistance to cisplatin [54]. Considering that these and other studies have shown that ZEB1 contributes to tumor aggressiveness through EMT-dependent and independent mechanisms [18], it would be interesting to determine if SPARC, through integrin αvβ3 and ZEB1, regulates other important aspects of tumor progression, such as the generation of cell populations with CSC phenotypes or therapy resistance.

## 4. Materials and Methods

### 4.1. Cell Lines

Cell lines were purchased from the American Type Culture Collection (ATCC, Manassas, VA, USA). LNCaP clone FGC (CRL-1740) cells and 22RV1 (CRL-2505) cells were maintained in Roswell Park Memorial Institute (RPMI) 1640 media (Gibco, Life Technologies, Carlsbad, CA, USA). PC3 (CRL1435) and DU145 (HTB-81) cells were maintained in Dulbecco’s Modified Eagle Medium (DMEM) F12 media (Gibco, Carlsbad, CA, USA). Both culture media were supplemented with 10% fetal bovine serum (FBS; Mediatech, Manassas, VA, USA), streptomycin–penicillin, and amphotericin B (Corning Inc., Corning, NY, USA). All cells used in this work were cultured at 37 °C in a humidified atmosphere with 5% CO_2_.

### 4.2. Lentiviral Transduction

PC3 cells with stable SPARC knockdown and LNCaP cells overexpressing SPARC were obtained as described previously [6]. DU145 cells with stable SPARC knockdown were obtained through transduction with lentiviral vectors containing a short hairpin RNA (shRNA) against human SPARC (pLenti-U6-shRNA [SPARC]-Rsv[GFP-Puro]). DU145 cells transfected with scramble shRNA (pLenti-U6-shRNA [Neg-control]-Rsv[GFP-Puro]) were used as the control. LNCaP cells with a stable knockdown of ZEB1 were obtained through transduction with lentiviral vectors containing a shRNA against human ZEB1 (pLenti-U6-shRNA [h ZEB1]-Rsv[GFP-Puro]). LNCaP cells transfected with scramble shRNA (pLenti-U6-shRNA [Neg-control]-Rsv[GFP-Puro]) were used as the control. All the lentiviruses used in this work were purchased from Gen Target Inc. (San Diego, CA, USA) and cells were transfected using a standard procedure. Briefly, 7.5 × 104 cells per well were seeded in 6-well plates. After 24 h, cells were incubated with lentiviral particles at a multiplicity of infection of 3, plus 5 μg/mL polybrene (Sigma-Aldrich, St. Louis, MO, USA) in 1 mL of culture media for 24 h. Later, cells integrating the vectors were selected using 2 μg/mL puromycin (Sigma-Aldrich, St. Louis, MO, USA) for 24 h.

### 4.3. Protein and Peptides

Recombinant human SPARC protein (Novus Biological, Centennial, CO, USA) was used to stimulate PCa cells at different concentrations. To block the activation of integrin αvβ3, cells were incubated with an Arginine–Glycine–Aspartic acid (RGD) peptide with strong affinity and selectivity for the integrin αvβ3 (ab142698, Abcam, Waltham, MA, USA).

### 4.4. Western Blot

Whole-cell protein was extracted using a radioimmunoprecipitation assay (RIPA) buffer supplemented with a protease inhibitor cocktail (Roche, Indianapolis, IN, USA). For the determination of phosphorylated proteins, the extraction buffer was supplemented with a phosphatase inhibitor cocktail (Cell Signaling, Danvers, MA, USA). Then, 10 to 50 μg of protein were loaded in 10% polyacrylamide gels, separated by SDS-PAGE, and transferred to a nitrocellulose membrane. Membranes were blocked with 5% BSA in Tris-buffered saline (TBS) with 0.2% Tween and incubated for 12 hrs at 4 °C with primary antibodies diluted in blocking buffer. After washing, bound primary antibodies were detected with secondary antibodies conjugated with horseradish peroxidase (HRP) and revealed with an enhanced chemiluminescence detection kit for HRP (EZ-ECL, Biological Industries, Cromwell, CT, USA). The antibodies used in this work are listed in Table 1.

### 4.5. Indirect Immunofluorescence

Cells were seeded on 12 mm coverslips at 60% confluence. After 24 h, cells were fixed in 4% paraformaldehyde for 30 min, permeabilized with 0.1% Triton X-100 in PBS for 10 min, washed, blocked with 3% BSA in PBS for 30 min, and incubated for 12 h with the primary antibody diluted in blocking buffer. Afterward, cells were washed in PBS and incubated with the secondary antibody diluted in blocking buffer for 45 min in a dark chamber. Finally, cells were washed in PBS, nuclei were stained with 4’, 6-diamidino-2-fenilindol (DAPI; 1:10,000, sc3598, Santa Cruz, CA, USA) and the coverslips were mounted in glass slides using anti-fade fluorescence mounting media. Primary and secondary antibodies used for this work are listed in Table 2.

### 4.6. RNA Extraction and RT-qPCR

Total RNA was extracted from cultured cells using TRIzol (Ambion, Life Technologies, Carlsbad, CA, USA) following standard procedures. RNA concentration was quantified using a BioTek Synergy HT plate reader (BioTek, Winooski, VT, USA) and 3000 nanograms of cDNA were synthesized using the kit cDNA Affinity Script qPCR (Agilent Technologies, Santa Clara, CA, USA). Afterwards, 100 ng of cDNA was amplified by qPCR using the kit Brilliant II SYBR Green qPCR Master Mix (Agilent Technologies, Santa Clara, CA, USA). The housekeeping gene pumilio RNA-binding family member 1 (PUM1) was used as a normalizer and the results were analyzed using the ΔΔCt method. The primer sets used for the qPCRs are detailed in Table 3.

### 4.7. Focal Adhesion Formation Assay

Focal adhesions analyses were performed as described [55]. Fifty thousand cells per well were seeded in 24-well plates on glass coverslips coated with fibronectin. After 24 h, cells were depleted from serum for 3 h. Subsequently, cell migration was induced with DMEM F12 medium supplemented with 10% FBS for 30 min. Cells were fixed with 4% w/v formaldehyde (158127; MilliporeSigma, Burlington, MA, USA) and 4% w/v sucrose (S0389; MilliporeSigma, Burlington, MA, USA) in Dulbecco’s PBS (DPBS), pH 7.4. Cells were permeabilized with 0.1% v/v Triton X-100, blocked in 4% w/v nonfat dry milk in DPBS for 30 min at 21 °C, and immunostained using the procedure described above. The primary and secondary antibodies used for this assay are listed in Table 2. The actin cytoskeleton was labeled using Phalloidin coupled to Alexa Fluor 488 (1: 5000, A12379, Invitrogen, Carlsbad, CA, USA). Nuclei were stained with Hoechst 33258 at 200 ng/mL (H3569; Thermo Fisher Scientific). Coverslips were washed three times with 0.1% v/v Triton X-100/DPBS and mounted with ProLong Gold (P36930; Thermo Fisher Scientific, Waltham, MA, USA). Images were acquired with a monochrome camera (CM3-U3-31S4M-CS; FLIR Systems) installed in an Eclipse TIU2 microscope (Nikon Instruments Inc., Melville, NY, USA) with a 60X oil immersion objective. Analyses of number/cell and size of focal adhesions were performed using Fiji ImageJ software (https://imagej.net/contribute/citing, accessed on 2 March 2022) [56]. 

### 4.8. Transwell Migration Assay

For the transwell migration assay, 5 × 104 cells per well were seeded in the upper chamber of a 96-well CytoSelect™ (Cell Biolabs, San Diego, CA, USA) plate with 8-μm pore membranes. Cells in the upper chamber were kept in culture media without FBS, whereas in the lower chamber, culture media with 10% FBS was placed as a chemoattractant. After 24 h, transmigrated cells were resuspended and dyed with CyQuant^®^ GR Dye (Cell Biolabs). Fluorescence at 485/528 nm was quantified in a BioTek Synergy HT plate reader (BioTek, Winooski, VT, USA).

### 4.9. Wound Healing Assay

Cells were seeded in 24-well plates at a density of 4 × 10^5^ and cultured in RPMI 1640 media supplemented with 10% FBS. After 24 h, cells were starved, and a scratch was made with a pipette tip. Cells were washed and incubated with RPMI 1640 media supplemented with 2% FBS and SPARC or SPARC plus RGD peptide were added to the media. The wound was photographed and the wound area closure was quantified using Fiji ImageJ software [56].

### 4.10. Statistical Analysis

Data analysis was performed with the GraphPad Prism 8 program (GraphPad Software, La Jolla, CA, USA). For all experiments, the data are expressed as mean ± standard deviation of at least three independent experiments. For two-group comparisons, the Mann–Whitney U test was used. For more than three groups, Kruskal–Wallis test was used to analyze the differences between the groups. In all cases, *p* ≤ 0.05 was considered statistically significant. The *p*-value, the number of experiments, and the statistical test used in each case are detailed in the legend of each figure.

## 5. Conclusions

Our data show that SPARC induces E-cadherin repression and enhances cell migration through integrin αvβ3 and the transcription factor ZEB1 in PCa cells. Knocking down SPARC in PCa cells increases E-cadherin expression, decreases FAK Y925 phosphorylation, impairs focal adhesion formation, and decreases the expression of the EMT regulator ZEB1 (Figure 6). It should be considered as the limitation of the cell line used.

## Figures and Tables

**Figure 1 ijms-23-05874-f001:**
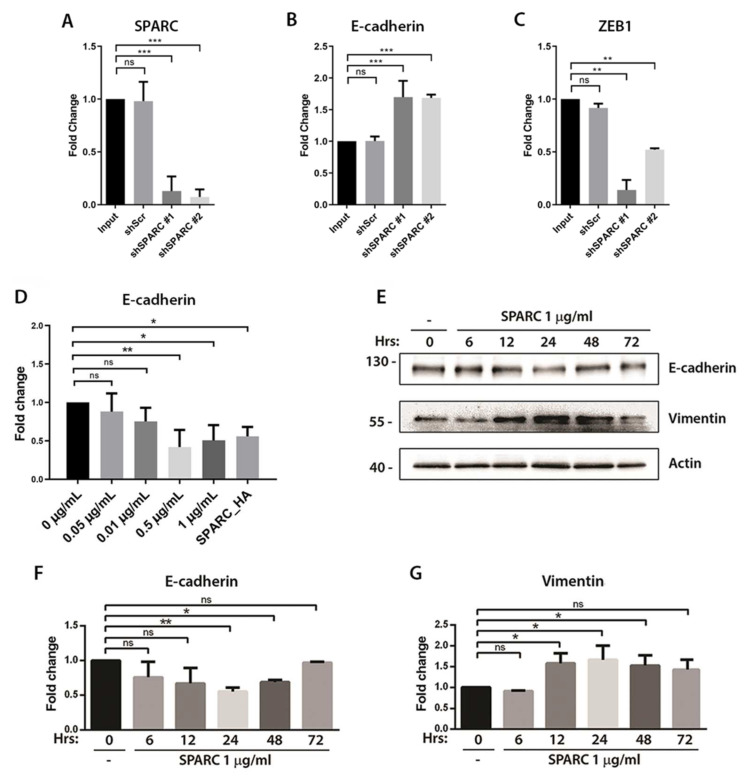
Effect of SPARC on E-cadherin, vimentin, and ZEB1 expression in prostate cancer cells (**A**–**C**). Relative expression of SPARC, E-cadherin, and ZEB1 in the prostate cancer cell line DU145 transduced with a lentiviral vector carrying a short hairpin RNA against SPARC (shSPARC) or a scrambled sequence (shScr) (**D**). Relative expression of E-cadherin in the prostate cancer cell line LNCaP treated with different concentrations of SPARC protein or transduced with a lentiviral vector carrying the sequence for SPARC (SPARC_HA) (**A**–**D**). Relative expression was normalized to pumilio and control cells (first column) using the ΔΔCt method (**E**). Representative Western blot of E-cadherin and vimentin in LNCaP cells treated with 1μg/mL of SPARC protein for up to 72 h (**F**,**G**). Quantification of the optic density of the Western blot shown in (**F**). Expression of E-cadherin and vimentin was normalized to β-actin (**A**–**D**,**F**,**G**). Data are expressed as mean ± SD (*n* = 3). ns = *p* > 0.05; * = *p* ≤ 0.05; ** = *p* ≤ 0.01; *** = *p* ≤ 0.001; Kruskal–Wallis test.

**Figure 2 ijms-23-05874-f002:**
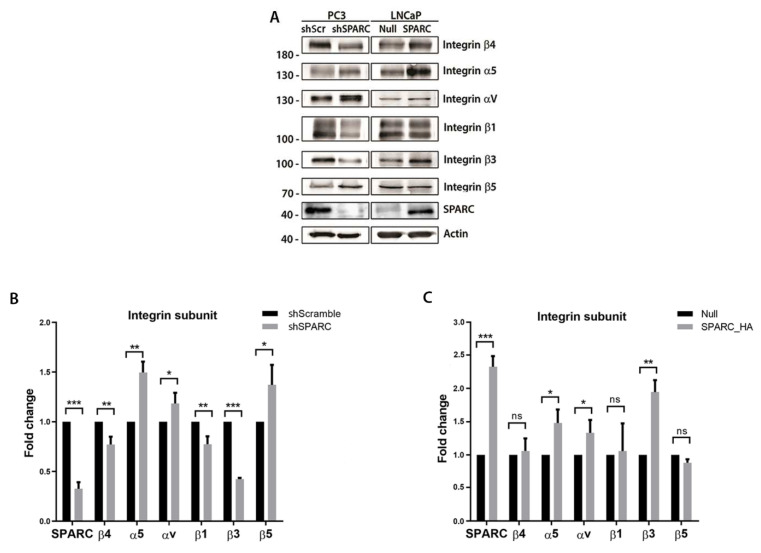
Effect of SPARC on the expression of integrin subunits and the inhibition of E-cadherin (**A**). Representative Western blot of integrin subunits and SPARC expression in PC3 cells transduced with a lentiviral vector carrying a short hairpin RNA against SPARC (shSPARC) or a scrambled sequence (shScr) (left panel) and in LNCaP cells transduced with a lentiviral vector carrying the SPARC sequence (SPARC_HA) or a control vector (null) (right panel) (**B**,**C**). Quantification of the optic density of the Western blot shown in (**A**). Expression of integrin subunits and SPARC was normalized to β-actin and control cells (**B**,**C**). Data are expressed as mean ± SD (*n* = 3). ns = *p* > 0.05; * = *p* ≤ 0.05; ** = *p* ≤ 0.01; *** = *p* ≤ 0.001; Mann–Whitney U test.

**Figure 3 ijms-23-05874-f003:**
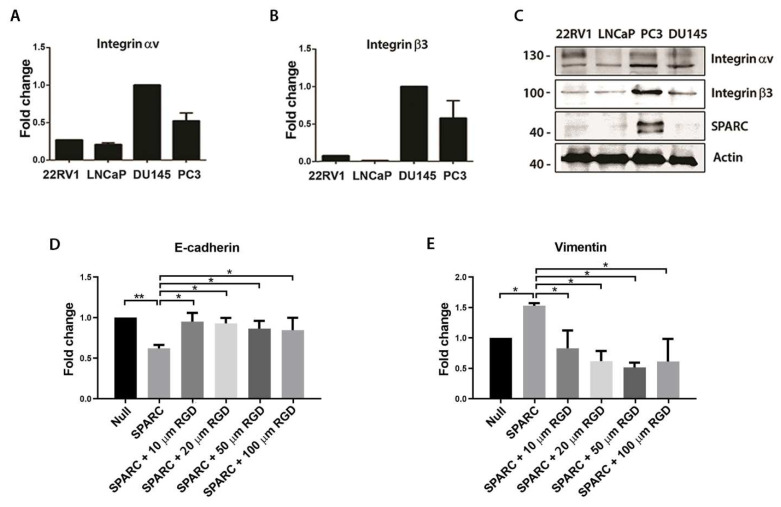
Expression of integrin αvβ3 subunits in prostate cancer cell lines and its effect on the inhibition of E-cadherin (**A**,**B**). Relative expression of integrin αv (left) and β3 (right) subunits measured by RT-qPCR in the prostate cancer cell lines 22RV1, LNCaP, PC3 and DU145 (**C**). Representative Western blot of integrin αv, β3 subunits and SPARC in different prostate cancer cell lines (**D**,**E**). Relative expression of E-cadherin (**D**) and vimentin measured by RT-qPCR in the prostate cancer cell LNCaP. LNCaP cells overexpressing SPARC were incubated with different concentrations of RGD peptide (**A**,**B**,**D**,**E**). Relative expression was normalized to pumilio and control cells (first column) using the ΔΔCt method. Data are expressed as mean ± SD (*n* = 3). ns = *p* > 0.05; * = *p* ≤ 0.05; ** = *p* ≤ 0.01; Kruskal–Wallis test.

**Figure 4 ijms-23-05874-f004:**
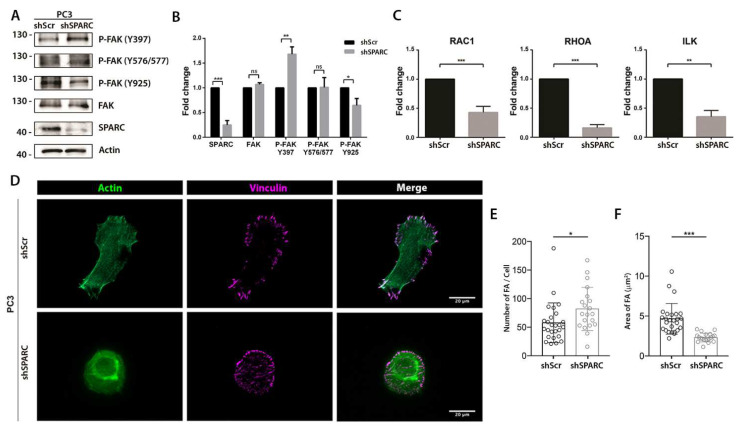
Effect of SPARC knockdown on the phosphorylation status of focal adhesion kinase (FAK) and the formation of focal adhesions (**A**). Representative Western blot of FAK, phosphorylated FAK (P-FAK) and SPARC expression in PC3 cells transduced with a lentiviral vector carrying a short hairpin RNA against SPARC (shSPARC) or a scrambled sequence (shScr) (**B**). Quantification of the optic density of the Western blot shown in (**A**). Protein expression was normalized to β-actin and control cells (**C**). Relative expression of RAC1, RHOA, and ILK measured by RT-qPCR in PC3 shScr and PC3 shSPARC cells. Relative expression was normalized to pumilio and control cells (first column) using the ΔΔCt method (**D**). Representative images of immunofluorescence vinculin and cytoskeleton stain with phalloidin in PC3 shScr and PC3 shSPARC cells. Bar = 20 μm (**E**). Quantification of the number of focal adhesions (FA) per cell in both conditions (**F**). Quantification of the area of FA in both conditions (**B**,**C**,**E**,**F**). Data are expressed as mean ± SD (*n* = 3). ns = *p* > 0.05; * = *p* ≤ 0.05; ** = *p* ≤ 0.01; *** = *p* ≤ 0.001; Mann–Whitney U test.

**Figure 5 ijms-23-05874-f005:**
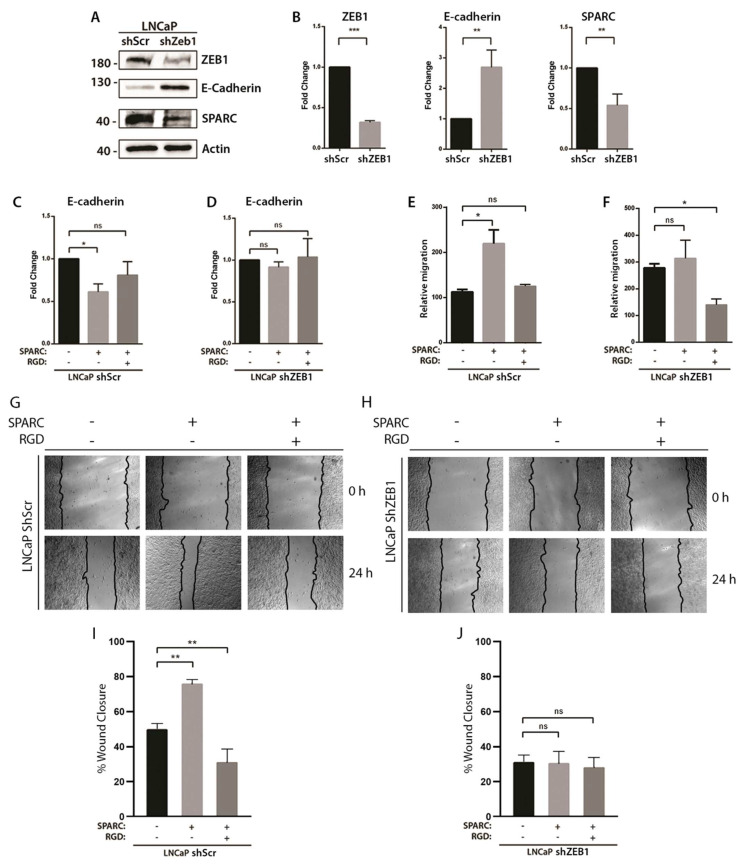
Effect of integrin αvβ3 blockade and ZEB1 knockdown on the SPARC-induced E-cadherin downregulation and enhanced migration (**A**). Representative Western blot of ZEB1, E-cadherin and SPARC in LNCaP cells transduced with a lentiviral vector carrying a short hairpin RNA against ZEB1 (shZEB1) or a scrambled sequence (shScr) (**B**). Quantification of the optic density of the Western blot shown in (**A**). Protein expression normalized to β-actin and control cells (shSCR) (**C**,**D**). Relative expression of E-cadherin measured by RT-qPCR in LNCaP shScramble (**C**) and LNCaP shZEB1 cells stimulated with 1 μg/mL SPARC and 50 μM RGD for 6 h. E-cadherin expression normalized to pumilio and control cells (first column) using the ΔΔCt method (**E**,**F**). Transwell migration assay of LNCaP shScramble (**E**) and LNCaP shZEB1 (**F**) cells stimulated with 1 μg/mL SPARC and 50 μM RGD for 24 h (**G**,**H**). Representative images of wound healing assay of LNCaP shScramble (**G**) and LNCaP shZEB1 (**H**) cells stimulated with 1 μg/mL SPARC and 50 μM RGD for 24 h (**I**,**J**). Quantification of the percentage of the wound area covered after 24 h (**B**–**F**,**I**,**J**). Relative migrations were compared with their own control (basal condition) (**B**–**F**). Data are expressed as mean ± SD (*n* = 3). ns = *p* > 0.05; * = *p* ≤ 0.05; ** = *p* ≤ 0.01; *** = *p* ≤ 0.001; (**B**)—Mann–Whitney U test; (**C**–**F**,**I**,**J**)—Kruskal–Wallis test.

**Figure 6 ijms-23-05874-f006:**
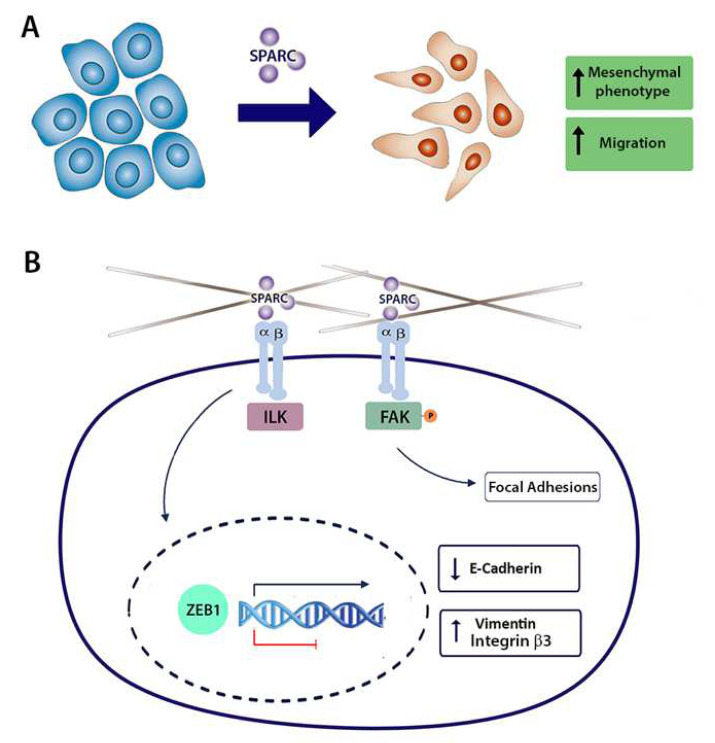
Proposed model for the effect of SPARC on E-cadherin expression and migration mediated by integrin α_v_β_3_ and the transcription factor ZEB1 in prostate cancer cells. (**A**) SPARC induces functional changes associated with the epithelial–mesenchymal transition (EMT): increased mesenchymal phenotype and enhanced motility. (**B**) Because SPARC requires the activity of integrin α_v_β_3_ to induce E-cadherin downregulation and cell migration, SPARC effect could be a result of the direct or indirect association of SPARC and the integrin α_v_β_3_. Through integrin α_v_β_3_, SPARC could activate the focal adhesion kinase (FAK) and the integrin-linked kinase (ILK). FAK activation by phosphorylation on Y925 promotes focal adhesion turnover, enhancing motility. On the other hand, through FAK, ILK, or other downstream molecules, SPARC could be promoting the expression of the EMT transcription factor ZEB1. The transcription factor ZEB1 inhibits the expression of E-cadherin and induces the expression of mesenchymal markers such as vimentin.

**Table 1 ijms-23-05874-t001:** Antibodies used for Western blot.

Antibody	Brand	Catalogue	Dilution
SPARC	Cell Signaling	5420	1:1000
ZEB1	eBioscience	14974182	1:1000
E-cadherin	BD Transduction Laboratories	610181	1:1000
Vimentin	AbCam	Ab8978	1:2000
Integrin β_1_	Cell Signaling	9669T	1:1000
Integrin β_3_	Cell Signaling	13166T	1:1000
Integrin β_4_	Cell Signaling	14803T	1:1000
Integrin β_5_	Cell Signaling	3629T	1:1000
Integrin α_5_	Cell Signaling	4705T	1:1000
Integrin α_v_	Cell Signaling	4711T	1:1000
FAK	Cell Signaling	13009T	1:1000
P-FAK (Y397)	Cell Signaling	8556T	1:1000
P-FAK (Y576/577)	Cell Signaling	3281T	1:1000
P-FAK (Y925)	Cell Signaling	3284T	1:1000
Actin	MP Biomedicals	691002	1:5000
Goat anti-Rabbit IgG HRP	Jackson Immunoresearch	111-035-003	1:10,000
Goat anti-Mouse IgG HRP	Jackson Immunoresearch	115-035-003	1:10,000

Abbreviations: SPARC—secreted protein acidic and rich in cysteine; ZEB1—zinc finger E-box-binding homeobox 1; FAK—focal adhesion kinase; HRP—horseradish peroxidase.

**Table 2 ijms-23-05874-t002:** Antibodies used for immunofluorescence.

Antibody	Brand	Catalogue	Dilution
Vinculin	Sigma-Aldrich	V9131	1:400
Donkey anti-Rabbit IgG Alexa Fluor 594	Life Technologies	A21207	1:500
Goat anti-Mouse IgG Alexa Fluor 555	AbCam	ab150118	1:500

**Table 3 ijms-23-05874-t003:** Sequence of oligonucleotides used as primers for the RT-qPCR.

Gen Name	Forward Primer	Reverse Primer
Secreted Protein Acidic and Rich in Cysteine (SPARC)	5′-AAC CGA AGA GGA GGT GGT-3′	5′-GCA AAG AAG TGG CAG GAA GA-3′
Cadherin 1 (E-Cadherin)	5′-GAA CGC ATT GCC ACA TAC AC-3′	5′-ATT CGG GCT TGT TGT CAT TC-3′
Vimentin	5′-GCC AAG GCA AGT CGC G-3′	5′-CAT TTC ACG CAT CTG GCG-3′
Zinc Finger E-Box Binding Homeobox 1 (ZEB1)	5′-TTC ACA GTG GAG AGA AGC CA-3′	5′-GCC TGG TGA TGC TGA AAG AG-3′
Integrin Subunit Alpha V	5′-TCT CTC GGG ACT CCT GCT AC-3′	5′-CTG GGT GGT GTT TGC TTT GG-3′
Integrin Subunit Beta 3	5′-ACC AGT AAC CTG CGG ATT GG-3′	5′-CTC ATT GAA GCG GGT CAC CT-3′
Rac Family Small GTPase 1 (RAC1)	5′-TCC GCA AAC AGA TGT GTT CTT A-3′	5′-ATG GGA GTG TTG GGA CAG TG-3′
Ras Homolog Family Member A (RHOA)	5′-GGT GAT GGA GCC TGT GGA AA-3′	5′-TGT GTC CCA CAA AGC CAA CT-3′
Protein Tyrosine Kinase 2 (Focal adhesion kinase; FAK)	5′-CAG GGT CCG ATT GGA AAC CA-3′	5′-CTG AAG CTT GAC ACC CTC GT-3′
Integrin Linked Kinase (ILK)	5′-CTT CCC TGG ATC ACT CCA CAG-3′	5′-GGG AGA AGC CAT GAT CGT CC-3′
Pumilio RNA Binding Family Member 1 (PUM1)	5′-CGG TCG TCC TGA GGA TAA AA-3′	5′-CGT ACG TGA GGC GTG AGT AA-3’

## Data Availability

Data are available from the authors upon request.

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
