# Peer review of "SPARC Induces E-Cadherin Repression and Enhances Cell Migration through Integrin αvβ3 and the Transcription Factor ZEB1 in Prostate Cancer Cells"

_ijms, 2022, doi:10.3390/ijms23115874_

Round 1

Reviewer 1 Report

Title: "Secreted protein acidic and rich in cysteine induces E-cadherin repression and enhances cell migration through integrin αvβ3 and the transcription factor ZEB1 in prostate cancer cells” 

Authors: Fernanda López-Moncada, María Jose Torres, Boris Lavanderos, Oscar

Cerda, Enrique A. Castellón, Héctor R. Contreras

Comments:

Integrins are known to mediate activation of the EMT program in the tumor microenvironment. In this work, Fernanda López-Moncada et al., the authors hypothesize that secreted acidic and cysteine-rich protein (SPARC) causes the suppression of E-cadherin through the activation of integrins and ZEB1. Interesting topic; however, there are some mistakes of carelessness:

Major points:

1: The title is long and confusing, perhaps SPARC or Osteonectin could be chosen instead of the full name.

2: The paper contains many complicated abbreviations due to the cell changes. Therefore, a list of abbreviations would be helpful for the inexperienced reader.

3: Page 3, Figure 1 E-G: Why was the transcription factor ZEB1 not also searched for in the Western blot?

4: Page 5, line 143 and p. 9, line 320: Here it is emphasized in each case that RGD highly specifically blocks αVβ₃-integrin. I think the authors should mention that RGD blocks different integrin subtypes and discuss what impact this fact has on their results.

5: Page 5, Figure 3 D, E: 

          - Again, why was the transcription factor ZEB1 not also searched for?

          - On what basis were the RGD concentrations chosen? They seem to me to be very high. For comparison, there are papers that use 1, 2 or 5 µM or even   500 nmol/l?

6: Page 6, Figure 4 D:  The immunofluorescence images should be larger and clearer. Also, results from at least two cell types should be shown and compared.

7: I do not understand why different cell types were chosen for the experiments, e.g., Figure 1 A-C (DU145), Figure 2A (PC3), etc. To demonstrate reproducibility, all results should be shown on at least two different PCa cell types.

8: Page 8, lines 272-273, Discussion: here the authors describe that integrin activity leads to an increase in Snail and Slug and in the abstract it is emphasized that SPARC increases ß3 integrin. This is in contradiction with the fact that at the same time the authors emphasize that SPARC does not lead to an increase in Snail and Slug. They refer to a previous work of their own (ref. 6) and do not elaborate in the present paper. I would like to see a short, recent study (e.g., using Western blot) that clearly rules this out, and in the discussion I miss a presumed rationale for the cause of this phenomenon.

9: Page 8, line 281, Discussion: please check the formatting of the references in the parentheses.

10: Minor formatting should be adjusted in the paper:

          -Page 7, line 236 Check sentence (of).

          -Page 8, line 281, Discussion (references in parentheses).

          -Page 12, line 405, legend Figure 6 (adhesion).

          -"Figure" and its corresponding letter should be capitalized throughout the paper (e.g., page 3, line 103, Figure 1E-G, etc.).

          -Some unnecessary spaces can be removed (e.g., page 1, line 39; page 2, lines 77 + 80, etc.).

Author Response

Answers to the Reviewer 1

Reviewer 2 Report

In this manuscript, the authors investigated the role of SPARC in E cadherin expression and cell migration. 

In general, this manuscript describes sound data. Nonetheless I have some concerns and comments regarding this manuscript. Mainly about choice of cell lines and missing functional data for EMT. 

In the introduction, I missed a statement about the consensus that the EMT process is not a binary switch that shunts from fully epithelial to fully mesenchymal extremes. Cellular plasticity should be explained. 

My major concern regarding the manuscript is that although solid experiments have been performed on molecular changed in various EMT markers, changes in cellular properties are much less investigated (only 1 migration assay in 1 cell line is shown). At least a migration and an invasion assay should be performed. Preferably for more PCa cell lines to show functional changes in the cells. 

Moroever, EMT programs are known to be linked to additional traits such as stemness and or survival rates. It would strengthen the manuscript to check at least one of those traits in the PCa cell lines. 

One other concern is the use of the PC3 cell line in several of the figures. Why did the authors choose this cell line? SPARC is known to be abundant in the bone marrow microenvironment and plays an important role in the deposition and assembly of collagen. Moreover, it also contributes to fibrotic diseases. It is known that prostate cancer evokes merely osteoblastic lesions in the bone, suggesting that SPARC might be involved in these processes. Why did the authors choose the PC3 cell line which results in lytic lesions in the bone? Why not performing these experiments in the VCaP cell line, also a metastatic cell line that expresses high levels of SPARC? This cell line would be much more relevant to the clinical situation.  

In the discussion, I missed a clear description when in the metastatic cascade the expression of SPARC is important for the migration. Is it only important for bone metastasis or also for other metastatic sites? Moreover, the statement that "showing that SPARC could be contributing not only to dissemination but also to the development of PCa" is not  investigated in this manuscript and not shown by the data presented. 

Conclusions:

"these results suggest that SPARC, through integrin avb3 and ZEB1 could also regulate other key processes for PCa tumor progression" this is too much speculation and not shown in your data.  

Author Response

Answers to the Reviewer 2

Reviewer 3 Report

In the proposed manuscript, Fernanda López-Moncada et al. report that SPARC-induced transitions between epithelial and mesenchymal states are mediated by integrin αvß3 and FAK signaling. Although described in more depth in their previous work, the EMT-like changes in the proposed manuscript are mostly documented by changes in E-cadherin and vimentin transcript abundance which is not sufficient. Although SPARC shRNA/OE induced effects on EMT markers are mostly consistent and statistically significant, the magnitude of changes is rather small. A functional experiment demonstrating changes in motile properties of the cells is necessary to support the change of phenotype.

In figures 2 and 3, the authors show a positive correlation between SPARC and integrin ß3 expression, but in follow up experiments, an integrin inhibitor with rather broad specificity was used. The role of integrin αvß3 in SPARC-induced EMT should be verified by a more specific approach and supported by a functional migration, invasion or motility experiment.

In figure 4, PC3 cells after SPARC silencing manifested changes in FAK signaling. Considering that the SPARC-induced EMT-like changes are ZEB1 dependent, which molecules are implicated in the signaling pathway linking the integrin – ILK or integrin – FAK to the modulation of ZEB1 transcriptional activity?

In the migration experiment with ZEB1 shRNA LNCaP cells in figure 5, ZEB1 knockdown apparently increases relative cell migration. This observation does not support the notion of increased migration of cells with mesenchymal phenotype. Can the authors comment this discrepancy?

Author Response

Answers to the Reviewer 3

Round 2

Reviewer 1 Report

Title: "Secreted protein acidic and rich in cysteine induces E-cadherin repression and enhances cell migration through integrin αvβ3 and the transcription factor ZEB1 in prostate cancer cells” 

Authors: Fernanda López-Moncada, María Jose Torres, Boris Lavanderos, Oscar Cerda, Enrique A. Castellón, Héctor R. Contreras

Comments:

Overall, the authors took the issues/criticisms raised seriously and argued some of them only in the letter but did not implement them in the paper.

There are still some important points:

1) The authors used different cell types throughout. For question 6 and 7, they always chose the most appropriate one for each experiment. i think 2 varieties of the cells should be shown everywhere. and if that is not possible, the limited transferability/conclusion of the study should be mentioned in detail in the discussion/conclusion.

2) I miss here that the specific role of integrin αvβ3 in SPARC-promoted conversion of cells from mesenchymal to epithelial form should definitely be demonstrated by a more relevant and practical migration experiment.

3) Again, I am missing which αvβ3 integrin-FAK-associated specific signaling pathways are responsible for the activation of the transcription factor ZEB1.

4) The authors show in Figure 5 that silencing of ZEB1 significantly and unexpectedly increases cell migration. How can we understand this discrepancy?

Minor points:

There must not be a period at the end of the title.

Correct page number, it says 7/12 everywhere.

Discussion, line 269: correct αvβ3.

Author Response

Answers to the Reviewer 1

Title: "Secreted protein acidic and rich in cysteine induces E-cadherin repression and enhances cell migration through integrin αvβ3 and the transcription factor ZEB1 in prostate cancer cells”

Authors: Fernanda López-Moncada, María Jose Torres, Boris Lavanderos, Oscar Cerda, Enrique A. Castellón, Héctor R. Contreras.

There are still some important points:

1) The authors used different cell types throughout. For question 6 and 7, they always chose the most appropriate one for each experiment. i think 2 varieties of the cells should be shown everywhere. and if that is not possible, the limited transferability/conclusion of the study should be mentioned in detail in the discussion/conclusion.

Answer: We thank again the reviewer point. Indeed, we used different cell types depending on what was the gene to be modified. We used DU145 cells when a regulating effect of SPARC on E-Cadherin was analyzed, due to in previous work, we described that the DU145 PCa cell line expresses low level of E-cadherin in comparison to other prostate cell lines. On the other hand, in other experiments the experiments we used PC3 cell line since this line has high levels of SPARC, thus a more clear effects are shown when silenced. But we agree with the reviewer and have included this point in the revised discussion.

2) I miss here that the specific role of integrin αvβ3 in SPARC-promoted conversion of cells from mesenchymal to epithelial form should definitely be demonstrated by a more relevant and practical migration experiment.

Answer: We agree with the reviewer comment. Finally, we have carried out functional migration experiment which results are included in the revised Figure 5. Also corresponding description of these assays was included in the revised Material and Method section and commented I revised discussion.

3) Again, I am missing which αvβ3 integrin-FAK-associated specific signaling pathways are responsible for the activation of the transcription factor ZEB1.

Answer: We thank the reviewer comment. Indeed, the scope of this study does not allow determining which of these pathways is involved. There is evidence in the literature that both ILK and FAK can induce the activation of transcription factors such as snail and Slug. However, there are no studies regarding ZEB1. Some of this is already incorporated into the discussion and we reinforced this point in the revised discussion.

4) The authors show in Figure 5 that silencing of ZEB1 significantly and unexpectedly increases cell migration. How can we understand this discrepancy?

Answer: We thank the reviewer comment. Actually, in each experiment, the results are normalized with an arbitrary value assigned to the control, so it should not be compared values between controls. On the other hand, the wound closure experiment (Figure 5G-5J) confirms that cells with ZEB1 silencing are indeed less migratory. We clarified this point in the revised Materials and Methods section.

Minor points:

-There must not be a period at the end of the title.

Answer: Thanks for the point. It was corrected.

-Correct page number, it says 7/12 everywhere.

Answer: We thank the reviewer. Page numbers were already in the frame sheet provided by the journal. For clarity we have erased the 7/12 numbers in all pages.

-Discussion, line 269: correct αvβ3.

Answer: Thanks for the point. It was corrected.

Reviewer 2 Report

The authors have addresses many of the concerns adequately and hopefully will perform the VCaP experiments in the future.  

Author Response

Answers to the Reviewer 2

Title: "Secreted protein acidic and rich in cysteine induces E-cadherin repression and enhances cell migration through integrin αvβ3 and the transcription factor ZEB1 in prostate cancer cells”

Authors: Fernanda López-Moncada, María Jose Torres, Boris Lavanderos, Oscar Cerda, Enrique A. Castellón, Héctor R. Contreras.

Comment: The authors have addresses many of the concerns adequately and hopefully will perform the VCaP experiments in the future. 

Answer: We appreciate the reviewer comment. Indeed, the experiments with VCaP are being planned.

Reviewer 3 Report

Title: "SPARC induces E-cadherin repression and enhances cell migration through integrin αvβ3 and the transcription factor ZEB1 in prostate cancer cells”

Authors: Fernanda López-Moncada, María Jose Torres, Boris Lavanderos, Oscar Cerda, Enrique A. Castellón, Héctor R. Contreras.

Major point

In the revised version of manuscript “Secreted protein acidic and rich in cysteine induces E-cadherin repression and enhances cell migration through integrin αvβ3 and the transcription factor ZEB1 in prostate cancer cells.“  by Fernanda López-Moncada et al., the authors discussed potential biological and clinical significance of the SPARC -- αvβ3 - -ZEB1 signaling cascade based on their previous publications. Still, the experimental readout covering expression of E-cadherin and vimentin is not sufficient to support conclusions, and a functional motility experiment is necessary to demonstrate the biological relevance of their findings. This should be stressed out in the experimental model of LNCaP cells in which vimentin upregulation can be associated with other biological processes such as senescence.

Minor point

Line 48-49: please correct the formulation about changes in vimentin expression

Author Response

Answers to the Reviewer 3

Title: "Secreted protein acidic and rich in cysteine induces E-cadherin repression and enhances cell migration through integrin αvβ3 and the transcription factor ZEB1 in prostate cancer cells”

Authors: Fernanda López-Moncada, María Jose Torres, Boris Lavanderos, Oscar Cerda, Enrique A. Castellón, Héctor R. Contreras.

Major point:

In the revised version of manuscript “Secreted protein acidic and rich in cysteine induces E-cadherin repression and enhances cell migration through integrin αvβ3 and the transcription factor ZEB1 in prostate cancer cells.“  by Fernanda López-Moncada et al., the authors discussed potential biological and clinical significance of the SPARC -- αvβ3 - -ZEB1 signaling cascade based on their previous publications. Still, the experimental readout covering expression of E-cadherin and vimentin is not sufficient to support conclusions, and a functional motility experiment is necessary to demonstrate the biological relevance of their findings. This should be stressed out in the experimental model of LNCaP cells in which vimentin upregulation can be associated with other biological processes such as senescence.

Answer: We thank the comment and consider carrying out functional experiments. Indeed, wound-healing assays were performed and the results are included in the revised Figure 5.  Also corresponding description of these assays was included in the revised Material and Method section. In addition, these results support our conclusions and are commented in the revised discussion section.

Minor point:

Line 48-49: please correct the formulation about changes in vimentin expression

Answer: We appreciate the reviewer comment. We apology for missing the word “decreased” before vimentin. It was corrected.

Round 3

Reviewer 1 Report

After authors'revision, the manuscript results improved and, I have no further comments.

Reviewer 3 Report

Major point: 

In the revised version of the manuscript “Secreted protein acidic and rich in cysteine induces E-cadherin repression and enhances cell migration through integrin αvβ3 and the transcription factor ZEB1 in prostate cancer cells.“  by Fernanda López-Moncada et al., the authors provided additional experimental evidence demonstrating the effect of SPARC - integring signaling on the motility of LNCaP cells as biological readout of the observed changes in epithelial/mesenchymal phenotype. Although the experiment successfully confirms the findings from the transwell migration assay, analysing the motility of SPARC silenced PC3 cells or in other prostate cancer models in combination with the inhibition of integrin signaling would strengthen the biological relevance of the reported findings. 

Minor point

Previous version of figure 5 was not removed from the manuscript.